Rotating cell culture system-induced injectable self-assembled microtissues with epidermal stem cells for full-thickness skin repair

Zhang Min 1 2
Huang Meng 1 2
Dong Xixi 3
Wang Yibo 2
Zhang Luyue 1 2
Wang Zhaoxiang 1 2
Cao Junkai 2 gujun_nj@yeah.net
1 Medical School of Chinese PLA , Beijing , China
2 Department of Stomatology, The First Medical Center, Chinese PLA General Hospital , Beijing , China
3 Department of Stomatology, The Fifth Medical Center of PLA General Hospital , Beijing , China
Camacho-Villegas Tanya
Electronic publication date: 2024 Oct 31
Publication date: 2024
Volume: 12
Electronic Location ID: e18418
Received 2024 Mar 26; Accepted 2024 Oct 7
Copyright: © 2024 Zhang et al.
Copyright year: 2024
Copyright holder: Zhang et al.
License: This is an open access article distributed under the terms of the Creative Commons Attribution License, which permits unrestricted use, distribution, reproduction and adaptation in any medium and for any purpose provided that it is properly attributed. For attribution, the original author(s), title, publication source (PeerJ) and either DOI or URL of the article must be cited.
License URL: https://creativecommons.org/licenses/by/4.0/

Keywords: Full-thickness skin defects, Epidermal stem cells, Rotating cell culture system, Biodegradable microcarrier, Microtissues

Funding: National Key R&D Program of China 2020YFC2008900 This work was supported by National Key R&D Program of China (2020YFC2008900). The funders had no role in study design, data collection and analysis, decision to publish, or preparation of the manuscript.

==============================
Epidermal stem cells (EpSCs) are crucial for wound healing and tissue regeneration, and traditional culture methods often lead to their inactivation. It is urgent to increase the yield of high quality EpSCs. In this study, primary EpSCs were isolated and cultured in a serum-free, feeder-free culture system. EpSCs are then expanded in a dynamic 3D environment using a rotating cell culture system (RCCS) with biodegradable porous microcarriers (MC). Over a period of 14 days, the cells self-assembled into microtissues with superior cell proliferation compared to 3D static culture. Immunofluorescence and qPCR analyses consistently showed that the stemness of the 3D microtissues was preserved, especially the COL17A1 associated with anti-aging was highly expressed in RCCS induced microtissues. In vivo experiments demonstrated that the group treated with 3D microtissues loaded with EpSCs showed enhanced early wound healing, and the injectable 3D microtissues were more conducive to maintaining cell viability and differentiation potential. Our study provides valuable insights into the dynamic 3D culture of EpSCs and introduces an injectable therapy using 3D microtissues loaded with EpSCs, which provides a new and effective approach for cell delivery and offering a promising strategy for guiding the regeneration of full-thickness skin defects.

Introduction

Skin is the first barrier against external damage, but it is susceptible to suffering various injuries and cause full-thickness skin defects (Santema, Poyck & Ubbink, 2016). Epidermal stem cells (EpSCs) cannot only maintain tissue homeostasis by providing new cells to replace those lost or injured, but also dominate wound healing and epidermis regeneration (Sun et al., 2013). During the 1980s, mixed cell populations containing EpSCs were used to treat burns, resulting in improved wound healing (Brockmann et al., 2018). However, cultured autologous grafts also experience stem cell loss, and appropriate culture conditions for EpSCs are thought to prevent this during the preparation of autografts (Charruyer & Ghadially, 2009). Therefore, the use of sufficient quantities of high quality EpSCs is crucial for their clinical applications.

Compared to two-dimensional (2D) planar cultures, three-dimensional (3D) cell cultures have the advantages of better maintaining cell-cell interactions, preserving biological potential, simulating the 3D microenvironment, and the importance of full skin regeneration (Nanba et al., 2021; Zhang, Xu & Hu, 2021). Although some encouraging results have been achieved in the tissue-engineered skin, there are still some problems that prevent the widespread, such as the possible transmission of disease from animal-derived materials (Browne, Zeugolis & Pandit, 2013). A novel elastic GMP gelatin-based porous 3D microcarriers (MC), 3D TableTrix®, is packaged into ready-to-use tablets, which has been used for large-scale cultivation of mesenchymal stem cells (Yan et al., 2020). 3D TableTrix® with good processing technology and has obtained qualifications for FDA, DMF, and CDE pharmaceutical excipients, ensuring favorable quality and safety. Notably, the biodegradable properties of microcarriers can prevent the role of second trypsin in cell culture and applications, thereby making it more conducive to maintaining cell viability. However, 3D cultivation of EpSCs using 3D TableTrix® has not been explored.

The rotating cell culture system (RCCS) is a suspension cell culture device, manufactured and approved by National Aeronautics and Space Administration (NASA), to simulate microgravity conditions, enhance cell interactions, and significantly affect the EpSCs metabolism (Li et al., 2020). It has been shown that NASA-approved rotary bioreactor supports the proliferation of EpSCs and the formation of multi-layer 3D epidermal-like structures (Lei et al., 2011). 3D culture has proven to be a promising way to exert cell function. Therefore, we cultured primary EpSCs in a serum-free, feeder-free culture system first. Then the rotating cell culture bioreactor (RCCS) combined with 3D TableTrix® were carried out to explore the impact of 3D dynamic culture on epidermal stem cells proliferation. In addition, we also investigated the therapeutic potential of injecting 3D microstructures loaded with EpSCs for full-thickness skin defects.

Methods

Isolation and planar culture of epidermal stem cells

Primary human EpSCs were isolated from skin samples discarded after children circumcision at the PLA General Hospitawith some modifications (Aasen & Belmonte, 2010; Tjin, Chua & Tryggvason, 2020). This study was approved by the Ethics Committee of the PLA General Hospital (S2023-738-01), and informed consent of patients. Briefly, the epidermal sheets were separated with 0.5% dispase II (17105041; Gibco, Grand Island, NYA, US) overnight at 4 °C and digested into individual cells using 0.25% trypsin at 37 °C for 2–3 min, repeated 2–3 times. The single-cell suspension was resuspended in complete KGM-Gold medium (KGM, 00192060; Lonza, Basel, Switzerland), and 5 × 106 cells were seeded into T75 flasks coated with 100 μg/mL type IV collagen (C5533; Sigma Burlington, MA, USA). After that, the rapid adherence cells were cultured in KGM medium at 37 °C in 5% CO2. Cells at 70–80% confluence were treated with 0.25% trypsin for subculture, and passages 1–3 cells were used for subsequent experiments. The culture medium was replaced every 2 days and maintained at 37 °C in an incubator with a humidity of 5% CO2. The cells were photographed by microscope (Nikon, TI2-U, Tokyo, Japan).

Flow cytometry

The main EpSCs markers CK19 and Integrin-β1 were identified by flow cytometry. The cells were fixed with 4% paraformaldehyde, permeabilized with 0.1% Triton X-100, and blocked with normal goat serum. Then the cells were incubated with anti-CK19 (1:100 dilution) and anti-integrin-β1 (1:100 dilution) antibodies for 30 min and washed three times with PBS containing 1% FBS. The mouse IgG stained cells were used as isotype control to account for any non-specific binding, unstained cells were used as blank control. After that, the cells were incubated with the secondary antibody for 30 min in the dark. Finally, the cells were washed with PBS three times, re-suspended in PBS, and analyzed by flow cytometry (BD Biosciences, Franklin Lakes, NJ, USA).

Immunofluorescence staining

Immunofluorescence analysis was used to evaluate the expression of CK19, Integrin-β1, p63, K5, and K10. The cells were initially fixed with 4% paraformaldehyde for 30 min, then permeabilized with 0.1% Triton X-100 for another 30 min, and finally blocked with normal goat serum for 30 min at room temperature. After that, the cells and microtissues were subjected to overnight incubation at 4 °C with antibodies against CK19, Integrin-β1, p63, and K10 at a dilution of 1/200. After that, the goat anti-rabbit secondary antibody was incubated at a dilution of 1/1,000 in the dark at room temperature for another hour. Nuclei of the cells were counterstained with DAPI. Finally, the cells and microtissues were observed using laser confocal scanning microscopy (Nikon, Ti A1, TI2-U, Tokyo, Japan).

3D static culture (SC) in vitro

A tablet (20 mg) of 3D TableTrix® microcarriers (W01-200; CytoNiche Biotech, Beijing, China) was placed in each well of non-treated six-well plates, and an appropriate amount of PBS or deionized water was added to the gaps between the plates to ensure a moist environment within the culture plate. A total of 300 µL P1 single cell suspension containing 1.25 × 106, 2.5 × 106, 5 × 106, 1 × 107, 2 × 107 cells were seeded into a tablet of microcarrier. The microcarrier was completely infiltrated without any excess cell suspension flowing out and incubated at 37 °C and 5% CO2 for 2 h. A total of 10 mL fresh complete culture medium was added to each well and gently stirred to evenly disperse the microcarrier in the culture medium, and then placed it in a cell culture incubator for routine cultivation. A total of 80% of the culture medium was replaced within 24 h and changed every 2 days.

In vitro 3D dynamic culture induced by RCCS

A total of 300 µL of EpSCs suspension containing 2.5 × 106 and 5 × 106 cells was first seeded into a tablet of microcarriers for 24-h static culture, and then carefully transferred into a 10 mL high aspect ratio vessel (HARVTM, Synthecon, Houston, TX, USA) with 10 mL complete culture medium and no bubbles for cultivation. The vessels were installed on the RCCS and rotated counterclockwise at 5 rpm for 3 days. The speed was changed to 10 rpm for 4 days, and then increased to 15 rpm until the end of 14 days of cultivation. The culture medium was replaced every 2–3 days to observe cell growth every day, and photographs were taken on days 1, 7, and 14.

Cell harvesting and enumeration

A total of 200 µL of cell-laden microcarriers was aspirated into a non-treated 48-well plate. When the microcarriers were fully dissolved, the attached cells were counted using a hemocytometer. The cell attachment efficiency was calculated using the following formula:

Cellproliferationrate(%)=Attachmentefficiency(%)=attachedcells/initialseedingdensity×100.

The cell attachment efficiency requires calculating the number of cells under the same number of microcarriers. To ensure consistent MCs quantity for each sample, the required sample volumes for day 1, 3, 7, and 10 were calculated based on 200 µL on the first day of cultivation:

V1=200μLofMCsuspension

V3=10mL10mL−N×V1×V1

V7=10mL10mL−N×V3×V3

V10=10mL10mL−N×V7×V7,andsoon,

where, N is 3, and V1, V3, V7, and V10 is the Volume of each sample at day 1, 3, 7, and 10, respectively.

The cell population doubling (PD) and population doubling times (PDT) are calculated as follows:

Populationdoubling=LNNt/N0LN2,or

Populationdoubling=logNt/N0log2,or

Populationdoubling=3.32×(logNt−logN0)=3.32×logNtN0,

Populationdoublingstime=tlog2(N0/Nt)=lg2lgNt−lgN0×t

where, Nt is the cell count at harvest; N0 is the cell count at the time of seeding, and t is the cultivation duration (h).

Cell counting Kit 8 assay

The cell counting kit 8 (CCK 8) (Dojindo, Mashiki, Tabaru, Japan) was used to assess cell viability and proliferation. A total of 100 µL of the samples from SC and RCCS was pipetted into 96-well plates and washed three times with medium to ensure a consistent volume of 100 µL. After 5 min of sedimentation, half of the medium without microcarriers was discarded and 135 µL of medium and 15 µL of CCK-8 solution were added to each well. The plates were then placed in a 37 °C incubator for 1 h and the absorbance at 450 nm was measured using a microplate photometer (VersaMax, Emmett, ID, USA). In addition, cell viability assays were determined by a 24-h culture of cells from the three groups: blank, EpSCs and MC group. A total of 100 µL medium was added to the blank group; 1 × 104 cells/well was seeded into EpSCs group, and 1 × 104 cells and MCs was added to MC group. The formula is as follows:

Cellviability=ODm−BlODc−Bl×100

where, ODm is the OD value of microcarrier; Bl is the OD value of blank, and ODc is the OD value of EpSC.

Live/dead fluorescence staining of cells on microcarriers

Cell proliferation and attachment were detected by live/dead staining. A total of 100 µL samples were aspirated into a well of 96-well plate and washed once with 100 µL preheated warm PBS to remove the supernatant without microcarriers. A total of 0.5 µL Calcein AM and 2 µL propidium iodide (PI) (L3224; Invitrogen, Waltham, MA, USA) were mixed into 1 mL PBS to prepare working solution. A total of 100 µL working solution was then added to per well and incubated at 37 °C in the dark for 15 min. Finally, the samples were washed, resuspended in PBS, and imaged using a fluorescence microscope (Nikon TI2-U, Tokyo, Japan) and confocal microscope (Nikon Ti A1, Tokyo, Japan).

Scanning electron microscopy

A tablet of 3D TableTrix® microcarriers was dispersed in deionized water in a 6 mm dish for 30 min, frozen at −20 °C overnight and lyophilized for 24 h. When culturing cells via RCCS on days 1, 7, 10, and 14, portions of the samples were pipetted into a transwell 48-plate. The samples were immediately fixed with 2.5% glutaraldehyde for 1 h at room temperature and then further incubated overnight at 4 °C. The samples were rinsed three times with PBS and dehydrated with serial concentrations of ethanol (30%, 50%, 70%, 80% ethanol each for 10 min; 90%, 95%, 100% ethanol for 10 min twice). The samples were then frozen overnight at −20 °C and lyophilized for 24 h. Finally, the samples were coated with gold for 120 s and imaged using a scanning electron microscope (SU8000; Hitachi, Tokyo, Japan).

Real-time qPCR (RT-qPCR)

Real-time qPCR (RT-qPCR) was used to detect the gene expression of COL17A1, K5 and K14, Ki67, ITGA6 and ITGB1 (Lim et al., 2022), K10, K1 and involucrin (INL) in 2D, SC, and RCCS culture cell aggregations. More than 1 × 106 cells were dissolved in Trizol and stored in an ultra-low temperature refrigerator at −80 °C, using within 1 month. Total RNA was extracted using the TRIzol method and the concentration of RNA was detected using a spectrophotometer. According to the manufacturer’s recommendation, 1–2 µg total RNA was reversed into cDNA using the PrimeScriptTMRT Master Mix kit (Takara, Tokyo, Japan) and stored at −20 °C or 4 °C. A total of 2 μL diluted cDNA was used to amplify target gene by TB Green®FastqPCR Mix kit (Takara, Toyobo, Japan) in the Roche LightCycler® 96 real-time fluorescence quantitative PCR system and analyze it in LightCycler96_1.1.0.1320. Due to the stable relative expression quantity, GAPDH was served as an internal reference gene, and 2−ΔΔct method was used to analyze the relative quantitative of target genes. Three repeated wells were set up in each group. Primers used for RT-qPCR are listed in Table 1. 2−ΔΔct was calculated as follows:

AverageΔCt=Ct1+Ct2+Ct33,

ΔCt(Ctl)=Ct(targetgene)−Ct(housekeepinggene),

ΔCt(2DCtlgeometricaverage)=ΔCt(Ctl)1+ΔCt(Ctl)2+ΔCt(Ctl)33,

ΔΔCt(SC)=ΔCt(SC)−ΔCt(2DCtlgeometricaverage),

ΔΔCt(RCCS)=ΔCt(RCCS)−ΔCt(2DCtlgeometricaverage),

so,Foldgeneexpression=2−ΔΔct

Table 1 Primer sequence for RT-PCR.

Genes	Forward (5′-3′)	Reverse (5′-3′)	
GAPDH	AACAGCGACACCCACTCCTC	CATACCAGGAAATGAGCTTGACAA	
ITGA6	ACTGTAGCGTGAACGTGAACTGTG	AAGGCTCGCATGAGAATGTCCAAG	
ITGB1	TTGTGAAGCCAGCAACGGACAG	CAAGGCAGGTCTGACACATCTCAC	
K5	CAACCCACTAGTGCCTGGTT	GACACACTTGACTGGCGAGA	
K14	AGCAGCAGAACCAGGAGTACAAG	GGCGGTAGGTGGCGATCT	
K10	ATGAGCTGACCCTGACCAAG	TCACATCACCAGAGGGACACA	
K1	ATTTCTGAGCTGAATTCGTGTGATC	CTGATGGACTGCTGCAAGTT	
KI67	ACGCCTGGTTACTATCAAAAGG	CAGACCCATTTACTTGTGTTGGA	
COL17A1	AGCGGCTACATAAACTCAACTGG	CCGTCCTCTGGTTGAAGAAG	
INL	GACTGCTGTAAAGGGACTGCC	CATTCCCAGTTGCTCATCTCTC	

Repair of full-thickness skin defects in nude mice

The nude mice used in this study were housed in an animal receiving room that meets SPF standards. Nude mice were kept in IVC cages, with less than five mice per cage and an ambient temperature of 22 ± 3 °C. All animals were allowed to freely access water and food disinfected under high pressure. The Ethics Committee of Beijing View solid Biotechnology Co.LTD provided full approval for this research (Permit No. VS2126A 00009). A total of 7–8-week-old BALB/c nude mice were used to construct a full-thickness skin defect model. To alleviate the pain of animals, the mice were anesthetized with inhaled isoflurane (INH), and their back skin was disinfected with 75% alcohol. Then, each nude mouse was created a full thickness wound with a diameter of 1 cm on the back skin. The power analysis was used to calculate the sample size. A total of 24 nude mice were randomly divided into four groups (n = 6/group) by random number method: sham group (PBS), EpSCs group (EpSCs alone), MC group (microcarriers alone), and MC+EpSCs group (EpSCs-loaded microtissues). All injection sites were evenly distributed and a1 mL syringe and a No. 9 needle were used. The wound healing time was recorded and analyzed using Image J, and the residual wound area rate was calculated using the following formula:

Residualwoundarearate=AnA0×100%,

where, A0 is the initial wound area at day 0, and An is the wound area at day N. If the animals experience severe pain, discomfort, or irreversible damage in the experiment that cannot be recovered or relieved, euthanasia should be considered to alleviate their pain. The mice subjected to euthanasia were placed in a euthanasia chamber and injected with carbon dioxide. The vital signs of the mice were observed until there was no breathing and heartbeat, and the bodies were removed. The nude mice in each group were sacrificed on day 14, and wound tissues were collected for histological analysis.

Hematoxylin-eosin and Masson’s trichome staining

The fresh tissue samples (n = 6/group) were embedded in paraffin to perform hematoxylin-eosin (H&E) and Masson’s Trichome staining. The slices were dehydrated, fixed, stained and imaged using a slide scanner. ImageJ was used to measure the epidermal thickness and Masson’s Trichome staining was used to quantify collagen deposition.

Statistical analysis

Graphpad Prism software (version 9.1.1) was used for statistical analysis. All data were expressed as the mean ± SEM (standard deviation) of at least three independent experiments. Statistical differences were evaluated using ANOVA with Tukey’s multiple comparisons test. P < 0.05 is considered as significant difference; *indicates P < 0.05; **indicates P < 0.01; ***indicates P < 0.001, and ***indicates P < 0.0001.

Results

Isolation, planar culture and identification of primary EpSCs

The P0-P1cells were small and bright, and closely arranged with paved cobblestone-like growth (Fig. 1A). All these characteristics were in accordance with those of epidermal stem cells. Immunofluorescence results showed that approximately all isolated cells expressed high levels of CK19, integrin-β1, and p63 protein (Fig. 1B). Flow cytometry analysis of EpSCs specific markers showed that the positive expression rates of CK19 and integrin-β1 were 95.45% and 99.25%, respectively (Fig. 1C), indicating that the EpSCs specific markers were strongly expressed in the EpSCs. The cytological observation, flow cytometry, and immunofluorescence results indicated that the purified EpSCs has been successfully isolated from the human foreskin.

Figure 1 Identification of epidermal stem cells.

(A) Cell morphology under light microscope, the scale is 100 μm; (B) flow cytometry analysis of EpSCs markers; (C) immunofluorescence analysis of EpSCs markers, the scale is 100 μm.

Characterization of microcarriers and 3D static culture in vitro

The isolated EpSCs were seeded into MC for in vitro static 3D culture (Fig. 2A). These elastic microcarriers compacted into a ready-to-use tablet and rapidly expanded into a porous sphere in several seconds when encountering liquid with a diameter of 100–300 µm (Figs. 2B and 2C). There was no significant difference in cell activity between the 3D and 2D planar cultures (Fig. S1A), indicating favorable compatibility of MC. As the cell number increased, 2.5 × 106 cells and 5 × 106 cells proliferated gradually with culture, while 1 × 107 cells and 2 × 107 cells did not tend to proliferate (Fig. 2D). Within 24 h, after three time points, the cell adhesion rates of 2.5 × 106 cells and 5 × 106 cells were higher than the other three groups (Fig. 2E). Similarly, within 7 days of incubation, only 2.5 × 106 cells and 5 × 106 cells showed a significant increasing trend. From day 3 to day 7, even 2 × 107 cells had a negative growth (Fig. 2F). These results indicated that 2.5 × 106 cells and 5 × 106 cells may be the optimal concentrations for cell proliferation.

Figure 2 The characterization of 3D microcarriers and in vitro static 3D culture.

(A) Schematic illustration of cells seeded into 3D microcarriers; (B) The morphological characteristics of 3D microcarriers; (C) The diameter of microcarrier was 100–300 µm; (D) The live/dead fluorescence images, the living cells in green and dead cells in red,the scale is 200 μm; (E) The cell attachment rate of different concentrations; * indicates P < 0.05, ** indicates P < 0.01. (F) The cell numbers of the cell different concentrations the asterisk represents the comparison to day 1 at that concentration, * indicates P < 0.05, the pound represents the comparison to day 3 at that concentration, # indicates P < 0.05. The software and links used in this figure and Figs. 3, 6, and 7 includes Fiji (https://imagej.net/software/fiji/), GraphPad Prism9 (https://www.graphpad.com/), NIS-Elements ImagingSoftware (https://www.microscope.healthcare.nikon.com/products/software/nis-elements/viewer) and CaseViewer (https://www.3dhistech.com/solutions/caseviewer/).

In vitro 3D dynamic expansion induced by RCCS

In the subsequent experiments, static cultured cells served as a control to further compare the proliferation of 2.5 × 106 cells and 5 × 106 cells under RCCS dynamic culture (Fig. 3A). After 14 days of cultivation, 3D cell aggregations were clearly formed through extracellular matrix self-assembly, which could be observed by the naked eyes in the reactor (Fig. 3B). RCCS culture seeded with 2.5 × 106 cells had a higher PD and was faster PDT than the other two groups on day 7, 10 and 14, whereas 5 × 106 cells were faster than the SC in terms of PDT (Figs. 3C, 3D). Whether PD or PDT, RCCS culture far exceeded SC. 2.5 × 106 cells in RCCS had more proliferative space than 5 × 106 cells.

Figure 3 In vitro 3D dynamic amplification induced by RCCS.

(A) Picture of RCCS device; (B) The observation of cell microtissue; (C) Comparison of cell population doubling; (D) Comparison of cell population doubling time under SC culture and RCCS culture. * indicates P < 0.05, ** indicates P < 0.01, *** indicates P < 0.001, **** indicates P < 0.0001. The software and links used in this figure and Figs. 2, 6, and 7 includes Fiji (https://imagej.net/software/fiji/), GraphPad Prism9 (https://www.graphpad.com/), NIS-Elements ImagingSoftware (https://www.microscope.healthcare.nikon.com/products/software/nis-elements/viewer) and CaseViewer (https://www.3dhistech.com/solutions/caseviewer/).

The cell status and cell proliferation of microtissues in RCCS and static culture was detected using live/dead staining (Fig. 4A). From day 7, the cell aggregations in the RCCS appeared as 3D multicellular spheres, i.e., microtissues, whereas the cell aggregations in the SC were less evident. The OD value and cell attachment rate of 5 × 106 cells under RCCS culture peaked on day 10 and declined on day 14, while the OD value and cell attachment rate of 2.5 × 106 cells showed an increasing trend (Figs. 4B and 4C), which was the optimal concentration for cell proliferation. The cell attachment rate of RCCS seeded with 2.5 × 106 cells was significantly higher than that of SC (Fig. 4D). Moreover, at a concentration of 5 × 106 cells, the cell attachment rate and OD value of cells cultured using RCCS was always greater than that of SC (Figs. S1B and S1C). After dynamic cultivation of RCCS, the EpSCs loaded microtissuses was placed in a flat culture plate for routine cultivation, and microtissuses were able to migrate to the culture plate, indicating that microtissuses have favorable cell migration ability (Fig. S1D).

Figure 4 The cell status under static culture and RCCS culture.

(A) The live/dead detection of cell proliferation, the scale is 200 μm; (B) Comparison of cell numbers; (C) Comparison of cell attachment rate; (D) Comparison of cell attachment rate between static culture and RCCS culture. * indicates P < 0.05, ** indicates P < 0.01, *** indicates P < 0.001, **** indicates P < 0.0001.

Characterization and identification of microtissues induced by RCCS

The cell aggregations grew gradually from day 7 due to the dual effects of rapid cell proliferation phase and dynamic culture, and the largest cell aggregations appeared at day 14 in 2.5 × 106 cells (Figs. 4A and 5A). The SEM images intuitively display the cell adhesion and the morphology of 3D cell aggregations (Fig. 5B). The monolayer flat cells initially adhered on the surface and pores of MCs; one microcarrier was almost completely occupied at day 7 with two layers of cells appearing; many multiple layers of cells appeared until day 10, which was more pronounced on day 14 (Fig. 5B). The cells in these microtissues were small, round and polygonal, and they were connected to microcarriers one by one, which is the basis of formation of microtissues.

Figure 5 The characterization of microtissues induced by RCCS.

(A) Live/dead fluorescence detection of cell aggregations formation, the arrows refer to the microtissues, the scale is 200 μm; (B) SEM images showing the morphology of cell adhesion microcarrier.

Microtissues cultivation and formation processes are shown in Fig. 6A. In addition, we conducted laser confocal three-dimensional imaging of the microtissues (Fig. 6B). The RT-qPCR results showed that compared with 2D culture, the expression of COL17A1, epidermal basal cell markers K5 and K14, and proliferation marker Ki67 in the microtissues generated by RCCS were remarkably increased. The expression of epidermal stem cell markers ITGA6 and ITGB1, epidermal basal cell markers K14, proliferation marker Ki67, and COL17A1 in RCCS was significantly higher than that in SC, while the differentiation marker K1 was significantly lower than that in SC (Fig. 6C). The cellular immunofluorescence results showed that the epidermal stem cell markers p63, CK19, and epidermal basal cell marker K5 were continuously abundant expressed, whereas the differentiation marker K10 was rarely expressed (Fig. 6D).

Figure 6 The formation and verification of 3D microtissues.

(A) Schematic of self-assembly to form microtissues induced by RCCS; (B) Laser confocal 3D imaging of microtissues. (C) The marker gene expression of 3D microtissues; * indicates P < 0.05, ** indicates P < 0.01, *** indicates P < 0.001, ns indicates no significant difference; (D) The cellular immunofluorescence detection of 3D microtissues, the scale is 100 μm. The software and links used in this figure and Figs. 2, 3, and 7 includes Fiji (https://imagej.net/software/fiji/), GraphPad Prism9 (https://www.graphpad.com/), NIS-Elements ImagingSoftware (https://www.microscope.healthcare.nikon.com/products/software/nis-elements/viewer) and CaseViewer (https://www.3dhistech.com/solutions/caseviewer/).

Application of EpSCs-loaded microtissues in skin wound model

The wound healing status of each group is shown in Fig. 7A. On days 3, 7, and 10, the residual wound area in microtissues group was significantly smaller than that of the other groups, and smaller than the sham group on day 14. EpSCs group was smaller than MC and the sham group on day 10 post-wound (Fig. 7B). All groups recovered the stratified epithelial structure after complete wound closure, while the epithelium of the EpSCs-loaded microtissues group was thicker than the other groups (Figs. 7C and 7D). Meanwhile, Masson’s Trichome staining revealed that under the newly generated epidermis, the EpSCs-loaded microtissues groups detected more collagen deposition (blue collagen) than other groups (Figs. 7C and 7E).

Figure 7 Effects of MC+EpSCs on skin wound healing.

(A) Photographs of skin wound model; (B) The comparison of residual wound area between treatment groups; (C) H&E and Masson’s Trichome staining of each group; (D) The quantification comparison of epithelial thickness; (E) The quantification comparison of collagen deposition. The asterisk indicates comparison with the sham group, * indicates P < 0.05, ** indicates P < 0.01, *** indicates P < 0.001, **** indicates P < 0.0001, and the pound indicates comparison with the EpSCs alone group, # indicates P < 0.05; The software and links used in Figs. 2, 3, 6, and this figure includes Fiji (https://imagej.net/software/fiji/), GraphPad Prism9 (https://www.graphpad.com/), NIS-Elements ImagingSoftware (https://www.microscope.healthcare.nikon.com/products/software/nis-elements/viewer) and CaseViewer (https://www.3dhistech.com/solutions/caseviewer/).

Discussion

EpSCs are the most critical cells that dominate wound healing and epidermis regeneration (Sun et al., 2013). Moreover, maintaining the optimal microenvironment or establishing an ideal 3D tissue structure in vitro has always been an important topic in the field of tissue-engineered skin regeneration (Zhang, Xu & Hu, 2021). In this study, primary EpSCs were enriched by the rapid type IV collagen adhesion and cultured using a serum-free and feeder-free culture system. Degradable 3D microcarriers and RCCS were then used for the 3D dynamic suspension culture of EpSCs, and the cells gradually proliferated and self-assembled to form 3D microtissues. Both RT-qPCR and cell immunofluorescence demonstrated that EpSCs grown in microtissues still maintained the stemness. Finally, a full-thickness skin defects model showed that the microtissues groups had a better promoting effect on wound healing in the early stage.

Epidermal stem cells are a subset of cells located in the basal layer of the epidermis, with unlimited proliferation ability. Over 20 distinct types of keratin expression have been identified in epithelial tissues, with varying distributions across different skin epithelial cells. CK19 is expressed in the basal layer where epidermal stem cells are located. It is a widely used epidermal stem cell marker, which helps distinguish between stem cells and differentiated cells within the epithelium (Hu et al., 2023; Diao et al., 2019). During the directed differentiation of basal cells, there is a loss of expression of integrin-β1 until the cells have completely exited the basal layer (Suzuki et al., 2010). P63 is markedly expressed in the basal layer, which contains stem cells, while its expression is markedly down regulated in the differentiated basal layer (SanzRessel, Massone & Barbeito, 2021). Previous study have shown that CK19, integrin-β1, and p63 protein are highly expressed in EpSCs and can serve as reliable markers for isolating human EpSCs (Zhou et al., 2015; Li et al., 2014). Therefore, CK19, integrin-β1, and p63 were selected as biomarkers for EpSCs in this study. The immunofluorescence staining in this study showed that almost approximately all isolated cells expressed high levels of CK19, integrin-β1, and p63 protein. Furthermore, the flow cytometry analysis also demonstrated that the positive expression rates of CK19 and integrin-β1 were 95.45% and 99.25%, respectively, which indicating successful isolation of human EpSCs. It is worth noting that this study used a serum-free and feeder-free culture system for culturing EpSCs, which can reduce the possibility of microbial contamination by viruses, fungi, mycoplasma, and other microorganisms (Nakamura et al., 2016). Combining cytological observation, flow cytometry, and immunofluorescence, it is feasible to isolate primary cells and culture them in serum-free culture system to obtain highly purified EpSCs.

In traditional static culture systems, the uneven diffusion of air, nutrients, and metabolic wastes is not conducive to cell proliferation, while rotational culture can prevent cell sedimentation, create a suspension culture environment, and enhance cell interactions (Wang et al., 2021). In the RCCS system, the container and medium rotate in unison, thereby minimizing the relative motion between the two. The continuous rotation of the system maintains the cells in a state of constant free fall within the medium. The centrifugal force generated by rotation acts to counteract the force of gravity. This results in cells experiencing a microgravity environment that is almost weightless, effectively simulating the microgravity environment. This unique microgravity environment minimizes destructive shear forces inside the culture container, allowing for the formation of large cell aggregates (Mitteregger et al., 1999; Licata et al., 2023). We confirmed that the RCCS advanced the cell proliferation than SC through the results of CCK-8 and cell proliferation efficiency, as well as PD and PDT. This study manifested that RCCS cannot only significantly improve intercellular adhesion, but also resulted in greater microtissues loaded with EpSCs compared with SC. In addition, live/dead staining and SEM were performed to observe the morphology of cell adhesion, proliferation, and 3D microtissues loaded with EpSCs after 1, 7, 10, and 14 days of cultivation. It was found that EpSCs connected multiple microcarriers into complex and irregular 3D structures, which is an organic combination of scaffold materials and cells. It cannot only provide a reliable source of seed cells for tissue engineering, and regenerative medicine, but also can be used for in vitro drug screening and toxicology research. It is worth noting that 3D TableTrix® used for large-scale cultivation of mesenchymal stem cells, it has been proven to be a cost-effective, industrial-scale production of quality hMSCs or MSC-derived products. In the future, microcarriers may be licensed for industrial scale production of high-quality EpSCs. Therefore, 3D TableTrix® and RCCS is an effective and promising self-assembly microtissues method that can be used for 3D cultivation of various human tissues.

Different cultivation conditions will affect proliferation, differentiation, and other specific cellular reactions of cells, such as the release of bioactive factors, synthesis of the extracellular matrix, and cell-cell interactions (Li & Kilian, 2015). It has been reported that keratinocyte stem cells lose their stem cell markers rapidly when isolated from their niche, while in 3D culture, their stem cell markers were normally expressed (Vollmers et al., 2012). The study on hEpSCs also suggested that the RCCS may provide a favorable environment for inhibiting hEpSCs differentiation (Lei et al., 2011). EpSCs undergoes asymmetric division to generate stem cells and transient amplifying cells (TAC). As the differentiation varies, epidermal cells express different keratins. Epidermal stem cells express CK19; epidermal basal cells and TAC express K5 and K14 with oriented differentiation ability, and differentiated terminal cells express K1 and K10 without proliferation and differentiation ability (Tjin et al., 2018; Rice & Rompolas, 2020; Nie, Fu & Han, 2013). The qRT-PCR results showed that the expression of stemness markers in 3D microtissues cultured with RCCS was significantly higher than that of SC, indicating that RCCS contribute to the maintenance of stemness. The immunofluorescence results showed that p63, CK19, and K5 were all highly expressed, while K10 was rarely expressed, further indicating that they originated from stem cells. The expression of K5 and K14 also indicated the coexistence of stem cells and TAC in 3D microtissues. Combined with the increased expression of Ki67, the 3D microtissues cultured with RCCS were conducive to the proliferation of EpSCs and promoted the downstream differentiation of EpSCs, but still maintained the stemness and did not form terminal differentiation. Studies have shown that a high level of COL17A1 expression may signify higher potential or quality (Liu et al., 2019), indicating the 3D microtissues may create more abundant extracellular matrix and a favorable niche for EpSCs.

At the same time, injectable microtissues have been adopted in several animal disease models, demonstrating good tissue regeneration effects (Wang et al., 2022). Injection of human adipose-derived stem cells-laden gelatin microcryogels cell microgel has been applied to treat full-thickness skin defects, which can significantly accelerate the wound healing (Zeng et al., 2015). It has shown that maintaining the stemness of EpSCs can accelerate wound healing in a rat full-thickness defect model by activating the Notch pathway (Lin et al., 2021; Zhang et al., 2020). Here, EpSCs-loaded microtissues were accelerated to close wound in the early stage, which is consistent with the result of previous studies (Guo & Jahoda, 2009; Lin et al., 2021; Li et al., 2021). Our research shows that in the early stage of wound healing, the residual wound area of the EpSCs+MC group was smaller than that of only MC or EpSCs treatment. This indicateds that the EpSCs-loaded microtissues further accelerates early wound healing to prevent wound infection, especially for large wounds that are prone to infection and ulceration in the early stage (Wang et al., 2018). H&E staining examinations showed that the microtissues groups can form thick epithelium; the microcarriers can probably provide adequate living space such as scaffold for the cells within the microtissues to exert their differentiation potential and form a thicker epithelial structure (Wang et al., 2016). Masson’s Trichome staining turns collagen blue and is used to detect collagen content in tissues (Yang et al., 2022). In this study, Masson’s Trichome staining was performed on treated mice, and it was found that more collagen deposition in the EpSCs-loaded microtissues groups. Although more evaluations are needed to treat skin defects in the future, our preliminary results indicate that an EpSCs-loaded microtissue is a promising treatment method for the full-thickness skin defects. Moreover, direct injection therapy of the microtissues loaded with EpSCs can avoid the harm of second digestion, maximizes cell viability, and can be integrated into newly formed tissues. Interestingly, the expression of COL17A1 in RCCS was significantly higher than 2D culture and SC. There are also reports that COL17A1 can accelerate skin wound healing (Liu et al., 2019) and enhance re-epithelialization phenotype in transgenic mice (Jacków et al., 2016). We boldly speculated whether the Notch pathway and COL17A1 are involved in the mechanism of EpSCs-loaded microtissues accelerated wound healing, which will be the further direction for us to explore their mechanisms.

There are also several limitations in this study. First, wound healing is a complex and continuous process, and the full thickness skin defect model of mice cannot fully simulate the human wound healing process, such as the exclusive occurrence of keloids in humans (Limandjaja et al., 2020). However, animal models are more suitable for therapeutic testing. The results provide preliminarily validate that the injection therapy of 3D microtissues loaded with EpSCs can accelerate the wound healing in the early stage of the mouse model, which is a promising treatment method for the full-thickness skin defects. Second, the staining results showed that the microtissues group can form thicker epithelium, but further testing of the tissue quality formed after treatment is needed in the future. Third, although the underlying mechanism is still unclear, our results indicate high expression in COL17A1 microtissues. Further investigation is needed to determine which cytokines and signaling pathways are crucial in the microtissues of EpSCs-loaded accelerating wound healing, particularly Notch pathway and COL17A1.

Conclusions

The findings indicated that the combination of biodegradable microcarriers and RCCS culture was beneficial for the proliferation and self-assembly of EpSCs to form 3D microtissues while maintaining the stemness and improving cell viability. Injection therapy of 3D microtissues loaded with EpSCs further accelerated the early wound healing, resulting in thicker epithelium and more collagen deposition in a mouse model, which is a promising treatment method for full-thickness skin defects.

Supplemental Information

Supplemental Information 1 Raw data.

Supplemental Information 2 (A) The comparison of cell viability; (B) The comparison of cell attachment rate; (C) The comparison of OD value; (D) Cell migration ability after RCCS culture.

Supplemental Information 3 ARRIVE 2.0 Checklist.

Supplemental Information 4 MIQE Checklist.

Additional Information and Declarations

Competing Interests

Author Contributions

Human Ethics

Animal Ethics

Data Availability

The authors declare that they have no competing interests.

Min Zhang conceived and designed the experiments, performed the experiments, analyzed the data, prepared figures and/or tables, and approved the final draft.

Meng Huang performed the experiments, prepared figures and/or tables, and approved the final draft.

Xixi Dong performed the experiments, prepared figures and/or tables, and approved the final draft.

Yibo Wang performed the experiments, authored or reviewed drafts of the article, and approved the final draft.

Luyue Zhang performed the experiments, authored or reviewed drafts of the article, and approved the final draft.

Zhaoxiang Wang performed the experiments, authored or reviewed drafts of the article, and approved the final draft.

Junkai Cao conceived and designed the experiments, analyzed the data, authored or reviewed drafts of the article, and approved the final draft.

The following information was supplied relating to ethical approvals (i.e., approving body and any reference numbers):

The study was conducted in accordance with the Declaration of Helsinki and the PLA General Hospital granted ethical approval to carry out the study within its facilities (Ethical Application Ref: S2023-738-01)

The following information was supplied relating to ethical approvals (i.e., approving body and any reference numbers):

The Ethics Committee of Beijing Viewsolid Biotechnology Co.LTD provided full approval for this research (Permit No. VS2126A 00009)

The following information was supplied regarding data availability:

The raw measurements are available in the Supplemental File.

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
