# Peer review of "Rotating cell culture system-induced injectable self-assembled microtissues with epidermal stem cells for full-thickness skin repair"

_PeerJ, doi:10.7717/peerj.18418_

## Round 0.1 · original submission · Major Revisions

After carefully revising the manuscript titled ¨Rotating cell culture system-induced injectable self-assembled microtissues with epidermal stem cells for full-thickness skin repair¨, some crucial issues need attention before reconsidering your manuscript and considering that two reviewers have contradictory opinions of your manuscript.

Major revision:
Specifically, A more detailed description of the methods used in your study is necessary. This should include citations or all necessary information related to controls, concentrations of antibodies, number of samples, incubation times, and statistical validations.
- Please circumscribe the conclusion based only on the results obtained in the investigation. If authors consider it necessary can include or combine the conclusion and perspective section.
- In the discussion, please declare the limitations of the investigation and how those impact the findings.
- Describe or discuss the differences between the human vs. murine models' cicatrization. Do those differences have an impact on your results?
- Declare the microtissue's stability after the bioprocess and include a perspective view of this technology focusing on future human applications.
- Provide a point-by-point reviewer response document.

·

Basic reporting

This article has an appropriate background structure. However, it requires extensive revision in the details, such as spaces after punctuation or parentheses, such as those detected in lines 38, 40, 51, 66, 71, 72, 80, 107, 213, 236, 258, and 286. Additionally, the authors require putting the formulas in the appropriate format for publication. English requires minor grammatical structure corrections. The hypothesis with relevance and according to results.

Experimental design

Regarding the figures, it seems extremely relevant to me that the authors indicate or mark with a box what they want to point out for the readers, just as they do from Figure 6. Additionally, consider having the images in the pixels and format suitable for publication since the images provided are blurry when enlarged and do not allow details to be seen.

Validity of the findings

The original document does not tacitly define the space that the article fills. However, it presents during the conclusions what and how this article seeks to answer how structured 3D cultures serve to structure and address tissue dehiscence in superficial lesions.
The authors present an adequate structure, both methodologies and results in correlation, and the conclusions are offered according to the characteristics of an original article. I have other observations with the writing as long as it corrects what is indicated in the first section and is accepted for publication. publication

Additional comments

In my perspective the authors need to upgrade the image quality in all figures and resume the findings perhaps in a table to make a capitulation for the journal readers

Reviewer 2 ·

Basic reporting

The authors investigated an interesting approach to form epidermal microtissues, however, the conduction of experiments has a lot of space for improvement.

Experimental design

The experimental design lacks some essential controls. The methods are very poorly described which makes it impossible to be replicated by other researchers.

Validity of the findings

The findings may be misinterpretations as the starting material - isolated cells – was not properly characterised. In addition, the vivo data are an overestimation concerning the relevance of the effect size.

Additional comments

Line 39: The authors mentioned “animal-derived materials may lead to disease transmission”, but use a gelatin-based microcarrier, which is again animal-derived material. How does the present study address the problem or does it simply ignore it?
Line 48: Please provide a reference for the statement: “Previous research has demonstrated that the NASA-approved rotary bioreactor supports EpSC proliferation and enables the formation of multi-layered epidermal structures.”
Line 48: Why is it relevant whether something is “NASA-approved”? What should this mean?
Line 49: Please remove the statement “Additionally, combining 3D and rotating cell culture systems has been shown to promote the proliferation and differentiation potential of rat bone marrow mesenchymal stem cells (BMSCs)(Tang et al., 2017).”, because BMSCs are not within the scope of the study.
Line 63: Digestion for how long?
Line 70+: Please desribe in more detail how the cell stainings and flowcytometric analysis were performed, in particular for the non-surface marker p63. Eg provide how much antibody was used per staining. In addition, isotype control labelings are missing, the CK19 positivity could therefore be a false positive. Also include stainings for CK10 and CK14, the markers of differentiated keratinocytes, as negative control (https://www.ncbi.nlm.nih.gov/pmc/articles/PMC8193873/).
Line 74+: Please desribe in more detail how the cell stainings were performed.
Line 84: Please decribe how cell seeding onto microcarriers was performed.
Line 90-96: This section is not understandable, please explain better what should be expressed by these calculations.
Line 99+: Why are three different equations given for population doubling, and two different terms for population doubling time?
Line 108: 24-hour cultures of what?
Line 113: What is AM dye? Which concentrations of dyes were used?
Line 117: Please describe the method in more detail, which gradient concentrations were used?
Line 121: How much microcarrier is 1 “tablet” eg in grams?
Line 122: Which “RCCS culture vessel” was used?
Line 132: Where are the scans of the Northern Blots?
Line 133: How much is 1-2 µL total RNA in µg?
Line 137: What do the authors mean by “fixed relative expression”?
Line 138: Which pores do the authors mean in the statement: “Three repeated pores were set up in each group”?
Line 158: Which diamater has a “No. 9 needle”?
Line 175: The sentence “P< 0.05 (*), P< 0.01 (**), P<0.001 (***), and P< 0.0001 (****) were considered statistically significant” is nonsense, the authors should reflect on their knowledge on statistics and what a p-value means. A result is either statistically significant or not based on one significance cutoff, which is consensually p<0.05. The other numbers describe the exact value of a calculated p-value.
Line 180: Based on morphology it is not possible at all to conclude whether a cell is an epidermal “stem cell”, also normal differentiated keratinocytes have the same morphology.
Line 186-196: As a fixed amount of microcarrier was used, how would the results look like if 4 times as much microcarrier was used when seeding 2*10^7 cells compared to the 5*10^6 cell group?
Line 214: How does Supplementary Figure 1D provide information on “cell migration ability”? This is just a micrograph showing microtissues and single cells?
Line 217: How is microgravity simulated in the RCCS system?
Line 270: SEM is not able to reveal the “whole process of cell proliferation”.
Line 276: “maintaining the stemness of EpSCs” – in Figure 6, the authors clearly show elevated expression of CK10 and CK14, not CK19, also ITGA6 and ITGB1 were not different to static culture, which in summary indicates that the seeded cells adopt a differentiated keratinocyte phenotype instead maintaining an EpSC phenotype – if there was an EpSC phenotype at all, which is nowhere clearly demonstrated in the study. Therefore, what want the authors to express with this statement? Why did the authors not investigate Notch pathway?
Line 279: Which previous studies?
Line 280: The study does not contain an experiment that directly shows “more collagen deposition”, on which basis do the authors conclude this?
Line 281: Why is the thicker epithelium a consequence of “the differentiation potential of EpSCs”?
Line 287: Why can it be inferred that elevated COL17A1 expression would “enhance cell migration”? Elevated COL17A1 gene expression not necessarily means elevated COL17 protein production!
Line 289: Which direction is provided? And which mechanisms do the authors mean? Please refrain from making unspecific statements such as “The above provides a basis and direction for us to further explore the mechanism.”
Figure 7B: Although the EpSCs+MC group might be statistically significant, the overall comparison to the healing trends of other groups do not seem very substantial. Therefore, what is the real benefit for wound healing, with regard to the cost-benefit relation? Do the EpSCs+MC integrate into the newly formed tissue? Besides a thicker epithelium after 14 days, how is the tissue quality, is it more functional than in the other groups?
Line 290: Please completely rewrite the conclusions section to contain actual conclusions and not general statements such as “Besides, the mechanisms of enhanced wound healing are also worth further in-depth study.”, this is not a conclusion. Conclusion are derived from the actual findings from a study.

---

## Round 0.2 · Minor Revisions

Dear Authors:

This actual version improves the manuscript. Nevertheless, some issues remain before acceptance. Please include a point-by-point response to the reviewer and academic editor.

1. Clarify the selection of CK19 and Integrin-β1 as marked for flow cytometry; discuss if you select simple epithelial cytokeratins for cell characterization or skin cytokeratins and how this CK19 cytokeratin relates to foreskin epidermal stem cells.
2. Clarify how your selected epithelial cells are classified as such only with CK19 cytokeratins flow cytometer detection. Specifically, explain the connection between the CK19 cytokeratin and the selected epithelial cells.
3. It is important to improve the description of the RCCS system, as this will support your findings. Please support your description with appropriate literature.

Finally, the conclusions have improved.

Reviewer 2 ·

Basic reporting

The authors substantially increased the quality of the manuscript by adding a range of more detailed descriptions of methods and experiments as requested by the reviewer.

Experimental design

The authors substantially increased the quality of the manuscript by adding a range of more detailed descriptions of methods and experiments as requested by the reviewer.

Validity of the findings

In figure 1, flow cytometry was not improved by performing istoype control experiments for CK19 as well as CK10 or CK14 as requested by the reviewer. Without these data, the claim of having isolated epidermal stem cells is unsubstantiated, regardless of what is published elsewhere.The requested experiments need to be performed before a publication of the study can be considered.

Additional comments

The question on "How is microgravity simulated in the RCCS system?" remains unanswered. The authors added content, however, there is only information presented that the RCCS system simulates microgravity - still without explanation how this microgravity is implemented by the system. In case the culture system was used by NASA in zero gravity environment, then operating the system on earth in presence of normal gravity can not simulate microgravity unless the system generates forces that counteract gravity on earth. Please explain how microgravity is simulated/implemented here.

---

## Round 0.3 · accepted · Accept

The authors addressed the reviewer's and editor's suggestions. Now, the manuscript is understandable and methodologically corrected.